# Intact Fish Skin Graft vs. Standard of Care in Patients with Neuroischaemic Diabetic Foot Ulcers (KereFish Study): An International, Multicentre, Double-Blind, Randomised, Controlled Trial Study Design and Rationale

**DOI:** 10.3390/medicina58121775

**Published:** 2022-12-01

**Authors:** Dured Dardari, Louis Potier, Ariane Sultan, Maude Francois, Jocelyne M’Bemba, Benjamin Bouillet, Lucy Chaillous, Laurence Kessler, Aurelie Carlier, Abdulkader Jalek, Ayoub Sbaa, Laurent Orlando, Elise Bobony, Bruno Detournay, Hilmar Kjartansson, Ragna Bjorg Arsaelsdottir, Baldur Tumi Baldursson, Guillaume Charpentier

**Affiliations:** 1Diabetic Foot Unit, Centre Hospitalier sud Francilien Corbeil Essonnes, 91100 Corbeil-Essonnes, France; 2LBEPS, IRBA, Université Evry Paris Saclay, 91025 Evry, France; 3Diabetology Department, CHU Bichat—Claude Bernard, 75018 Paris, France; 4Institut Necker-Enfants Malades, Université Paris Cité, INSERM UMR-S1151, CNRS UMR-S8253, 75006 Paris, France; 5Diabetology Nutrition Department, CHU Montpelier, Université de Montpellier, 34090 Montpellier, France; 6Inserm, CNRS, Phymedexp, CHU de Montpellier, 34090 Montpellier, France; 7Diabetology Department, CHU Reims, 51100 Reims, France; 8Department of Diabetology, CHU Cochin, 75000 Paris, France; 9Department of Endocrinology-Diabetology, Dijon University Hospital, 21000 Dijon, France; 10INSERM Unit, LNC-UMR 1231, University of Burgundy, 21078 Dijon, France; 11Department of Endocrinology, Metabolic Diseases and Nutrition, University Hospital of Nante, 44000 Nantes, France; 12Department of Diabetology, CHU Strasbourg, 67000 Strasbourg, France; 13CERITD (Center for Study and Research for Improvement of the Treatment of Diabetes), Bioparc-Genopole Evry-Corbeil, 91042 Evry, France; 14CEMKA, 43 boulevard du Maréchal Joffre, 92340 Bourg-la-Reine, France; 15Landspitali University Hospital, 105 Reykjavik, Iceland

**Keywords:** diabetes foot ulcer, cell and/or tissue-based wound care products, intact fish skin graft

## Abstract

Background: Cell and/or tissue-based wound care products have slowly advanced in the treatment of non-healing ulcers, however, few studies have evaluated the effectiveness of these devices in the management of severe diabetic foot ulcers. Method: This study (KereFish) is part of a multi-national, multi-centre, randomised, controlled clinical investigation (Odin) with patients suffering from deep diabetic wounds, allowing peripheral artery disease as evaluated by an ankle brachial index equal or higher than 0.6. The study has parallel treatment groups: Group 1 treatment with Kerecis^®^ Omega3 Wound™ versus Group 2 treatment with standard of care. The primary objective is to test the hypothesis that a larger number of severe diabetic ulcers and amputation wounds, including those with moderate arterial disease, will heal in 16 weeks when treated with Kerecis^®^ Omega3 Wound™ than with standard of care. Conclusion: This study has received the ethics committee approval of each participating country. Inclusion of participants began in March 2020 and ended in July 2022. The first results will be presented in March 2023. The study is registered in ClinicalTrials.gov as Identifier: NCT04537520.

## 1. Introduction

While the prevalence of diabetes is increasing in developed countries, the quality of management of diabetes is improving and consequently, the cost of management is also increasing [1,2]. Rates of diabetes-related complications have declined substantially, but a large burden of the disease persists because of the continued increase in the prevalence of diabetes and cost of management of diabetes complications is also increasing [1,2,3]. Diabetic foot complications are serious and costly [4]. Diabetic foot ulcer (DFU) most often results from the combination of two major complications of diabetes: diabetic neuropathy and angiopathy, often complicated by soft tissue and bone infection. Arterial disease represents the most severe prognosis in terms of amputation and mortality. DFUs are a severe complication of diabetes mellitus and impact morbidity, mortality, and healthcare expenditure in a serious way. It is known that that 19–34% of patients with diabetes are likely to be affected with a DFU in their lifetime, and the International Diabetes Federation reports that 9.1–26.1 million people will develop DFUs annually [5]. Patients with DFUs were also found to have a 2.5-fold increased risk of death compared with their diabetic counterparts without foot wounds [6]. Treatment of DFUs accounts for approximately one-third of the total cost of diabetic care, which was estimated to be USD 176 billion in direct healthcare expenditures in 2012 [7]. Despite these high healthcare costs, about 20% of patients have unhealed DFUs at 1 year [8]. Even after wound resolution, subsequent DFUs are common, with a recurrence rate of roughly 40% of patients within 1 year [5]. Although there are well-established principles for managing DFUs, it is always a challenge. The fundamental care of a DFU includes sharp debridement, off-loading, and diabetic foot education in addition to local wound care with or without surgical debridement, dressings promoting a moist wound environment, wound off-loading, vascular assessment, general medical assessment, treatment of infections, and glycaemic control [9,10]. Thus, diabetic foot care must be multidisciplinary.

Cell and/or tissue based wound treatment products (CTPs) have slowly been advancing in the treatment of non-healing ulcers in the last 20 years. The first large studies on CTPs were published in 2005 for use on venous leg ulcers [11]. The largest of these studies used material from pig small intestine in the intervention group and later a number of other materials have been studied and used, mainly decellularized membranous organs from mammals but also cellularised skin equivalents and freeze dried amnionic membranes. In 2013, a new product was approved in the US by the Food and Drug Administration, decellularized fish skin [Kerecis^®^ Omega3 Wound™, Kerecis, Iceland]. Delivered as a sterilized, freeze-dried material, the fish skin graft has the benefit of not being treated with antibiotics and virus inactivating methods, thereby allowing the natural omega-3 fatty acids to remain, and it is a by-product of the food industry. Therefore, the fish derived CTP is both ecologically sustainable as well as rich in naturally occurring soluble molecules and omega-3 fatty acids. Omega-3 fats seem to have a multitude of positive actions, including an anti-inflammatory function and to some extent anti-bacterial properties. In clinical use it has promoted healing of chronic ulcers [12], consequently, its evaluation in the treatment of DFU is legitimate.

## 2. Materials and Methods

This study, KereFish, is the French part of a multinational clinical trial taking place in France, Germany, Italy, and Sweden. The overarching study, Odin, is a multi-national, multicentric, randomised, controlled, open-label, interventional study comparing the use of Kerecis Omega3 Wound with conventional therapy (SOC, Standard of Care) in the treatment of complex, hard-to-heal, diabetic foot wounds with a planned total of 330 patients.

Primary objective: To demonstrate the superiority of Kerecis Omega3 Wound over SOC in the treatment of severe diabetic wounds with or without a moderate arterial component.

Secondary objectives: The secondary objectives are to: (i) Evaluate the safety of Kerecis Omega3 Wound. (ii) Assess secondary clinical efficacy endpoints. (iii) Assess patient and caregiver satisfaction. (iv) Assess the impact on patient quality of life. (v) Assess the economic impact of Kerecis Omega3 Wound treatment on patient care costs and to determine the effectiveness of this treatment.

Primary endpoint: Percentage of healed wounds with complete epithelialisation at 16 weeks.

Secondary endpoints: Change in ulcer grade according to the University of Texas diabetic wound classification at each weekly visit. Change in quality of life. Change in pain. Healing trajectory. Cost effectiveness. Number of participants with fully healed ulcers at 20 weeks. Number of participants with fully healed ulcers at 24 weeks. Percentage of ulcers healed 50% or more at 12 weeks.

Inclusion criteria: People living with diabetes of any age with diabetic foot ulcer Grade 2 or 3 according to the University of Texas (UT) diabetic foot classification system [13]; Grade 2: wound penetrating to the tendon or capsule. UT Grade 3: wound penetrating to bone or joint, OR patients admitted/ambulatory for diabetic foot wounds or amputations, which have not closed or are dehiscent. Patients who can tolerate aggressive surgical debridement. Patients without severe ischaemia; ankle brachial index (ABI) > 0.6 or big toe pressure > 50 mmHg if ABI is not possible. Wound age ≥ 30 days (does not apply to amputation wounds; patients can be included when the wound is less than 30 days old) or if the amputation level is below the ankle. Patients willing and able to give informed consent to participate in the clinical trial. Male or female over 18 years of age. Patients living at a geographical distance compatible with the home nurse.

### 2.1. Conduct of the Study

This study uses the Smart-Trial^®^ electronic platform for gathering data, obtaining CRFs, and monitoring of the study. The study design is demonstrated in Figure 1. Randomisation will take place after inclusion via the electronic case report form. The randomisation list is in blocks of two; the stratification will be performed according to two criteria: amputation and non-amputation wounds and ABI (≤0.9 and >0.9). Patients will then be assigned either to the conventional treatment group (control group), or to the interventional group, where they will be treated with Kerecis Omega3 Wound. The hospital investigator will assess the wound status of patients in both groups during hospital visits at the following times:Week 7: patients come to the hospital for the investigator to assess wound progress.Week 16: end of study visit after the last application of Kerecis Omega3 Wound. An HbA1c measurement will be collected.Week 20: patients come to the hospital for the investigator to assess the progress of the wound.Week 24: at the end of the two-month follow-up period an HbA1c measurement will be collected.

The referral nurse (RN) and the home nurses (HNs) for the study in each centre will receive theoretical and practical training from a Kerecis professional specialized in wound care.

In the study, the difference between the standard of care group and the interventional group is that Kerecis Omega3 Wound is applied between the dressing, that would otherwise contact the ulcer, and the ulcer bed. The dressing on top of Kerecis Omega3 Wound then becomes the covering dressing. In the control group, the care is the same whilst there is no Kerecis Omega3 Wound under the covering dressing. Most often, the covering dressing would be a foam dressing. The different approaches for using offloading in different clinics will adhere to the International Working Group on the Diabetic Foot’s Guidelines [10].

Kerecis Omega3 Wound is applied once per week for the first six weeks, and every other week for the remaining 8 weeks of active intervention, with up to 10 applications in total. If complete epithelialization is achieved, the application will be stopped. Both groups will receive standard of care for the last two weeks (Weeks 15 and 16). For both groups each week, when the home nurse takes a picture of the wound, he/she will ask the patients to fill in a visual analogue scale (VAS) for perception of pain.

For the management of wounds in the conventional group, a specific care protocol will be communicated to those involved in the patient’s care. This protocol will include indications for the following elements of wound care: (i) Cleaning: basic hygiene care, Isotonic saline solution, running water and neutral soap. (ii) Debridement (aggressive, surgical) of the wound will be mechanical using a fixed blade scalpel/scalpel No. 10 or 15, or a sterile single-use curette such as MediSet^®^. The debridement procedure will be performed from the centre to the periphery of the lesion by removing non-vital components in the tangential planes and removing overhanging edges and hyperkeratosis and acanthosis of the skin at the wound edges. Mild bleeding attests to the quality of the debridement. (iii) Recovery: various dressings can be used: hydrocolloids, hydrofibres, alginate, hydrocellular and charcoal dressings, silver dressings, wound contact dressings, including fatty tulle (Vaseline gauze) and compression dressings (postoperative indication). Practical Guidelines on the prevention and management of diabetic foot disease of the International Working Group on the Diabetic Foot (IWGDF) are also used for choice of the local recovery [10].

### 2.2. Patient Monitoring

When epithelialisation is complete, treatment and visits from the home nurses will be stopped. Follow-up will be carried out by the referent nurses, who will visit the patient monthly or see him/her in hospital to check the wound, take a picture of the wound and collect information in case of a new hospitalization, relapse, or other important events until Week 24. The duration of the participation in the study: 16 weeks/patient + 8 weeks follow-up with a focus on wound healing at 16 weeks.

### 2.3. Photography

The ulcer is photographed before debridement/cleaning, after debridement/cleaning if that is deemed necessary, and after placement of Kerecis Omega3 Wound (if applicable) with a centimetre scale and patient identifier placed in the photo. An acetate tracing (outline of the ulcer on transparent plastic) is made and photographed where a centimetre scale and a patient identifier is visible in the photo. Photos are uploaded via the Smart-Trial^®^ portal as a part of filling out the case report form). The largest diameter, and the diameter perpendicular, is measured at all visits.

Three status checkpoints are imbedded in the study:i.Has the wound changed its grade, i.e., have the granulations covered bone and tendon? Answered by clinician since the answer is dependent on probing to bone.ii.Is the wound ready for standard of care only, to commence healing? Answered by clinician and blinded panel of clinicians.iii.Would a split thickness skin graft be a logical next intervention? Answered by blinded panel only.

### 2.4. Statistical Methods

The null and alternative hypotheses are:▪H0: no difference in the percentage of patients cured with complete epithelialization at 16 weeks between Kerecis Omega3 Wound and the SOC.▪H1: a percentage difference in the number of patients cured with complete epithelialization at 16 weeks between Kerecis Omega3 Wound and SOC.▪It is expected that 30% of patients will have healed with complete epithelialization at 16 weeks in the SOC group [see point below].▪A 20% improvement with Kerecis Omega3 Wound (i.e., 50% of patients healed) would be considered clinically significant.▪A Fisher’s exact conditional test with a two-sided significance level of 0.05 will have 80% power to detect the difference between a Group 1 proportion of 0.3 and a Group 2 proportion of 0.5 for a size d sample in each group of 90.▪Assuming an attrition rate of 10%, 180 patients (=190/0.9) will need to be randomized in this trial.

Statistical analysis will be performed using SAS^®^ software, version 9.4.0 or higher. Analysis of the primary endpoint (i.e., the percentage of patients healed with complete epithelialization at 16 weeks) will be conducted using Fisher’s exact test. The 95% confidence intervals of the proportions in each group will be displayed.

## 3. Discussion

In Explorer study [14], the sucrose octasulfate dressing significantly improved wound closure of neuroischaemic diabetic foot ulcers without affecting safety after 20 weeks of treatment along with standard care. Our study has a concept to replace the extracellular matrix (ECM) of chronic wounds, which is inflammatory in nature and cannot promote healing. Chronic wound fibroblasts cannot reorganize the ECM [15] and are insensitive to growth factors and other signalling factors [16], chronic wound fluid contains excessively high levels of metalloproteinases factors [17], partly due to inflammation. Fibronectin, a key component of provisional ECM that provides important binding sites is rapidly degraded by proteases [18]. In fact, growth factors themselves are also rapidly degraded by proteases [19]. Chronic wounds also lack the integrin receptor for fibronectin binding and keratinocyte migration [20]. The proteoglycan and glucosamine signalling of this construct is not yet well defined and may be less robust than the better characterized ECM properties of porcine small intestine (SIS) submucosa. This construct has been well characterized as containing proteoglycans and glycosaminoglycans, as well as endogenous active growth factors [21]. In hard-to-heal patients as defined above, we intend to try to save these hard-to-heal wounds with the intact fish skin graft observation wise, intact fish skin graft has the potential to move the wound from an inflammatory phase to a healing phase [22], we wish to use this graft in wounds that are difficult to heal. The technology provides a natural structure that contains proteins and fats (including omega-3) allowing stem cells and cells to migrate in the fish skin graft subsequently breaking it down to leave the cells and vessels invading the graft as a new foundation for healing.

It is estimated that two thirds of people with diabetes will develop peripheral neuropathy leading to loss of sensation, pain, and abnormal foot architecture [23], a quarter will develop a foot ulcer [24]. Of these, over half will become infected requiring hospitalization and a fifth of these will result in amputation [25]. It is clear that diabetes and its complications will place a greater burden on health care and financial resources and this situation is expected to worsen [26]. The reduction in ulcer size obtained after 4 weeks can predict the rate of ulcer healing, if the reduction is not close to 50% after 4 weeks, the ulcer has a chance of being healed between 9 and 30% only after 12 weeks [22]. Chronic ulcers are a port of invasion of bacteria and as noted above and are a marker for a high risk of amputation [24]. Rapid ulcer healing in this patient group is therefore needed, and there has been an upsurge of research into strategies to improve wound healing rates [27].

To our knowledge, the Explorer study [14] was the first to assess the efficacy of a dressing in individuals with DFUs associated with neuropathy and peripheral artery disease. The KereFish trial has the objective of investigating the reduction of the healing time of neurovascular ulcers in patients living with diabetes with an ABI lower than that taken into account in the Explorer Study [14] even if the devices used in this study are not part of the same category of wound treatment. The KereFish trial is also considered as the first study in therapy strategy considered passive in neurovascular lesions using skin graft [27]. A small series has demonstrated the potential for intact fish skin graft rich in omega-3 to accelerate wound healing in diabetic foot wounds and warrants further analysis in a randomised controlled trial, potentially as a routine adjunct in postoperative wound management, but those series had no control group which seems to be a principal limitation to prove the effectiveness of this type of therapy.

## 4. Conclusions

The aim of the KereFish and larger Odin study is to demonstrate the superiority of Kerecis Omega3 Wound over SOC in the treatment of diabetic wounds. 180 patients should be enrolled at the end of the inclusion period which is end of July 2022 and the first results are planned to be published in March 2023.

## Figures and Tables

**Figure 1 medicina-58-01775-f001:**
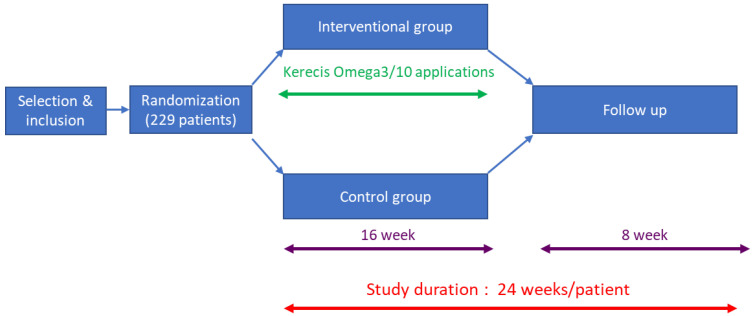
KereFish study design.

## Data Availability

All data are available in the manuscript.

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
