# Peer review of "Intact Fish Skin Graft vs. Standard of Care in Patients with Neuroischaemic Diabetic Foot Ulcers (KereFish Study): An International, Multicentre, Double-Blind, Randomised, Controlled Trial Study Design and Rationale"

_medicina, 2022, doi:10.3390/medicina58121775_

Round 1

Reviewer 1 Report

The authors present the aim and methodology of a multicentric, multinational study, KereFish, that investigates a novative approach in diabetic foot ulcer. The idea of the study is very interesting, I am wainting to see the next papers describing the results that were obtained.

As an observation, please detail the standard of care that was used for the patients included in the comparison group.

Author Response

As an observation, please detail the standard of care that was used for the patients included in the comparison group.

thank you very much for your comments, we have added a chapter detailing the standard of care, this action will be guided by the recommendations of the International Working Group of the Diabetic Foot

Reviewer 2 Report

The goal of this work is to test the hypothesis that a larger number of severe diabetic ulcers and amputation wounds, including those with moderate arterial disease, will heal in 16 weeks when treated with Omega3 Wound™ than with standard of care.

The design of the study is not appropriate and must be improved. In this work it is necessary to have more tests including histopathological investigations.

Author Response

many more thanks for your comment, we have established our hypothesis and the design of the study on an equal intervention in both groups, the role of the nurse referent will be to guarantee this equitability, The study has olso un  objective of using practices identical to the management of diabetic foot wounds in the real life , for this reason we have not included hystological sampling, this action is not retained as a recommendation by The International Working Group on the Diabetic Foot (IWGDF)